# Adjustment of Planned Surveying and Geodetic Networks Using Second-Order Nonlinear Programming Methods

**Murat Mustafin and Dmitry Bykasov \***

Department of Engineering Geodesy, Saint-Petersburg Mining University, 199106 Saint-Petersburg, Russia;
Mustafin_MG@pers.spmi.ru
* Correspondence: s195082@stud.spmi.ru

**Abstract:** Due to the huge amount of redundant data, the problem arises of finding a single integral solution that will satisfy numerous possible accuracy options. Mathematical processing of such measurements by traditional geodetic methods can take significant time and at the same time does not provide the required accuracy. This article discusses the application of nonlinear programming methods in the computational process for geodetic data. Thanks to the development of computer technology, a modern surveyor can solve new emerging production problems using nonlinear programming methods—preliminary computational experiments that allow evaluating the effectiveness of a particular method for solving a specific problem. The efficiency and performance comparison of various nonlinear programming methods in the course of trilateration network equalization on a plane is shown. An algorithm of the modified second-order Newton's method is proposed, based on the use of the matrix of second partial derivatives and the Powell and the Davis–Sven–Kempy (DSK) method in the computational process. The new method makes it possible to simplify the computational process, allows the user not to calculate the preliminary values of the determined parameters with high accuracy, since the use of this method makes it possible to expand the region of convergence of the problem solution.

**Keywords:** nonlinear programming; optimization methods; geodetic computations; Newton's method; trilateration network; conjugate gradient method

## 1. Introduction

Over the past thirty years, surveying and geodetic equipment has made a great leap forward. Such a rapid development of technology allowed surveyors to receive and process an enormous amount of data about objects. The use of devices, such as tacheometers, laser trackers and scanning laser systems, as well as satellites in surveying and geodetic practice, made it possible to increase the speed and accuracy of the data obtained. The use of modern surveying and geodetic methods in the construction of buildings is especially important; this is noted in works [1,2], as well as when determining deformations [3].

An important element in the solution of any surveying and geodetic problems is the office processing of measurement results (rejection of gross errors, equalization, assessment of the accuracy of the solutions obtained). Redundancy of measurements increases the accuracy and plausibility of the obtained solutions; however, as the amount of information obtained increases, the complexity of data processing also increases, as noted in articles [4,5]. There is a need to use computers with high performance characteristics in order to solve the problem in special software; this thesis is confirmed in the works of L.A. Goldobina [6], N.S. Kopylova [7], V.F. Kovyazina [8], A.M. Rybkina [9,10] et al. [11,12]. The development of computer technology makes it possible to automate the solution of many engineering problems, as well as to carry out computational experiments by modeling; in the works of authors, such as P.A. Demenkova. [13,14], N.V. Vasilyeva [15], E.V. Katuntsova [16] and A.A. Kochnevoy [17] et al. [18–20], special software products were used to solve engineering

problems. Nevertheless, the redundancy of measurements allows the surveyor to choose the optimal solution, taking into account the limiting criteria. However, in a situation where the data array is huge and the power of the computer does not allow quick processing and the obtaining of the result, the solution process can be optimized by various methods. In this regard, the topic of optimizing solutions for various industrial surveying and geodetic problems is very relevant; this idea finds its confirmation in articles [21–23].

In the mathematical aspect, optimization should be understood as a sequence of actions, the implementation of which contributes to obtaining a solution or clarifying an existing one. Optimization methods have been used for a long time in geodesy and surveying; the famous geodesist and mathematician K. Gauss is the author of many papers on this topic. There are many groups of optimization methods that can be applied in geodesy and surveying. It should be noted that the problems associated with solving nonlinear equations differ from linear problems in that there is no single, standard solution method. Depending on the restrictive conditions and the objective function type, a different set of solutions can be obtained, the best of which shall be chosen. Therefore, the study of the possibility of using various methods to solve problems of a certain group is the best way to choose an optimization method for solving a specific problem. The article discusses methods of nonlinear programming. For a number of reasons, these methods are best suited for their implementation in the surveying and geodetic computational process, namely:

(1) nonlinear programming methods allow the nonlinear and linear conditions that limit the objective function to be taken into account;
(2) these methods allow the solving of large systems of equations using algorithms that are most suitable for implementation on modern computers;
(3) using some nonlinear programming methods (such as second-order Newton's method) makes it possible to solve nonlinear equations without linearizing the original parametric equations;
(4) using nonlinear programming methods, it is possible to obtain a solution not only using the objective function of the least squares method, which is a classical method in geodesy and surveying, but also in other ways in accordance with the selected criterion function.

The third point is especially important, since there are a lot of problems in geodesy and surveying, where the desired parameters can be determined by solving nonlinear systems of equations, for example: calculating transition keys, equalizing surveying and geodetic networks and building terrain models. The fourth point makes it possible to experiment and choose other optimization criteria, different from the least squares method.

From all of the above, it can be concluded that it is advisable not only to apply nonlinear programming methods in surveying and geodetic computations, but also to improve their algorithms for geodesy and surveying in the future. Among the methods of nonlinear programming, two main groups can be distinguished—these are methods based on the use of derivatives of various orders and methods that calculate the extremum point without using derivatives (direct search methods). In this work, the methods of the first group are considered in detail, since their use provides a number of advantages:

(1) a large number of previously developed methods that have clearly formulated algorithms that are easy to implement with a computer;
(2) the ability to use several methods at once at different stages of solving one problem, in order to obtain the best result.

It should be said that the methods of this group have serious downsides. The main one is the preliminary preparation of the problem for the solution. It is necessary to calculate derivatives of different orders at each iteration; for this, an algorithm is drawn up in advance, according to which the derivatives will be calculated for a specific objective function. It takes a particularly long time if the function is not specified analytically, then it becomes necessary to calculate the derivatives by a numerical method. It is also

necessary to take into account that the objective function shall be continuous, otherwise the problem will have no solution. These downsides are reflected by G.G. Shevchenko in her works [24,25] and she proposes to use direct search methods (which do not use derivatives in the iterative process) when solving surveying and geodetic optimization problems.

The article analyzes the possibility of applying the second-order Newton's method, when solving surveying and geodetic optimization problems; in particular, when equalizing the surveying and geodetic network of trilateration on a plane. Today, design and equalization of surveying and geodetic constructions using new methods is a very relevant topic; this is confirmed in works [24–26]. Newton's method was chosen because it has the following upsides:

(1)　the method has a quadratic convergence rate of the iterative process, in contrast to first-order methods (gradient methods), which have a linear convergence rate;

(2)　for any quadratic objective function with a positive definite matrix of second partial derivatives (Hessian matrix), the method gives an exact solution in one iteration;

(3)　low sensitivity to the choice of preliminary values of the determined parameters, in comparison with gradient methods.

The second-order Newton's method was used to equalize the planned trilateration network.

## 2. Materials and Methods

### 2.1. Mathematical Justification for Solving the Task

The second-order Newton's method is included in the group of nonlinear programming methods of the second order [26,27]. More generally, the second-order Newton's method is an iterative method that applies a quadratic approximation to the original nonlinear objective function at each iteration. To evaluate the convergence of the method, a necessary condition is the threefold differentiability of the studied function. The existence of the second derivative at the extremum point provides a high rate of convergence of the method, in comparison with the first-order methods [28,29]. The method was studied in detail in the work of N.N. Eliseeva [30] and in the works [31–33], and was also applied by the authors of the article in [34]. However, the possibility of using the method in surveying and geodetic practice, when solving production problems, has almost not been studied. There were a number of objective reasons for this, which will be discussed below.

To derive the main formula of the second-order Newton's method, it is necessary to expand the original objective function in a Taylor series (1):

$$f(x) \approx f(x^{\otimes}) + f'(x^{\otimes}) \cdot (x - x^{\otimes}) + \frac{1}{2} f''(x^{\otimes}) \cdot (x - x^{\otimes})^2, \tag{1}$$

where $f'(x^{\otimes})$ is the first-order derivative with respect to the function $f(x^{\otimes})$, $x^{\otimes}$ is the minimum point of the function, $f''(x^{\otimes})$ is the matrix of the second derivatives of the objective function $f(x)$ in point $x^{\otimes}$.

The second-order Newton's method is based on the quadratic approximation of the function; therefore, the first three terms are taken into account in the Taylor series to derive the iterative formula [27,35]. Having received the value $x^{\otimes}$, it is possible to calculate the next approximation $x_{k+1}$ to the extremum point. Replacing in the Formula (1) $x^{\otimes}$ with $x_k$, and $x$ with $x_{k+1}$, also marking $\Delta x_k = x_{k+1} - x_k$, one can get the Formula (2):

$$f(x_{k+1}) \approx f(x_k) + f'(x_k) \cdot \Delta x_k + \frac{1}{2} f''(x_k) \cdot \Delta x_k^2. \tag{2}$$

To determine the extremum in the direction $\Delta x_k$, it is necessary to differentiate the function $f(x_{k+1})$ for each of the components $\Delta x_k$ and equate the resulting expression to zero (3):

$$f'(x_k) + f''(x_k) \cdot \Delta x_k = 0. \tag{3}$$

Expressing the variable $x_{k+1}$ from Formula (3), the main formula of the second-order Newton's method is obtained, according to which the iterative process (4) is constructed:

$$x_{k+1} = x_k - \frac{f'(x_k)}{f''(x_k)}, \tag{4}$$

where $f'(x_k)$ is the first derivative of the function $f(x)$ in point $x_k$ in the approximation $k$; $f''(x_k)$ is the second derivative of the function $f(x)$ in point $x_k$ in the approximation $k$.

Formula (4) describes an iterative process for a function of one variable. By writing expression (4) in matrix form (5), one can obtain an iterative formula of the method for the multidimensional case (functions of several variables):

$$X_{k+1} = X_k - H_k^{-1} \cdot \nabla f_k, \tag{5}$$

where $\nabla f_k$ is the column vector of the matrix of the first derivatives (gradient) of the objective function in the approximation $k$, $H_k$ is the matrix of the second partial derivatives (Hessian matrix) of the objective function with the target dimension of $n \times n$ in the approximation $k$; $X_k$ is the column vector of the determined parameters in the approximation $k$; $X_{k+1}$ is the column vector of the determined parameters in the approximation $k + 1$ [33].

A distinctive feature of the classical second-order Newton's method is that it is not necessary to determine the iteration step in the iterative process. The rate of convergence, as well as the direction of the search, depends on the Hessian matrix (6):

$$H(x_1, \ldots, x_n) = \begin{pmatrix} \frac{\partial^2 f(x^1,\ldots,x^n)}{\partial x^1 \partial x^1} & \frac{\partial^2 f(x^1,\ldots,x^n)}{\partial x^1 \partial x^2} & \cdots & \frac{\partial^2 f(x^1,\ldots,x^n)}{\partial x^1 \partial x^n} \\ \frac{\partial^2 f(x^1,\ldots,x^n)}{\partial x^2 \partial x^1} & \frac{\partial^2 f(x^1,\ldots,x^n)}{\partial x^2 \partial x^2} & \cdots & \frac{\partial^2 f(x^1,\ldots,x^n)}{\partial x^2 \partial x^n} \\ \cdots & \cdots & \ddots & \\ \frac{\partial^2 f(x^1,\ldots,x^n)}{\partial x^n \partial x^1} & \frac{\partial^2 f(x^1,\ldots,x^n)}{\partial x^n \partial x^2} & \cdots & \frac{\partial^2 f(x^1,\ldots,x^n)}{\partial x^n \partial x^n} \end{pmatrix}. \tag{6}$$

The main downside of the second-order Newton's method is the calculation of the Hessian matrix [27]; therefore, this method is little used in practice, since the calculation of the Hessian matrix at each iteration is a rather complicated computational process. However, the introduction of personal computers made it possible to automate the process of calculating the Hessian matrix. The calculation of partial derivatives can be implemented by a numerical method using one of the programming languages. Due to this, the problem of calculating partial derivatives for any objective function can be fully automated.

At each iteration, it is necessary to determine the sign of the Hessian matrix. The matrix of the second partial derivatives at each iteration shall be positive definite $H(f) > 0$; only if this condition is met, will the search direction lead to a decrease in the objective function $f(x)$. In iterations where the Hessian matrix is negatively defined, $H(f) < 0$, the direction of the search for the minimum target shall be replaced. In this work, it is proposed to use gradient methods at iterations where the matrix of second derivatives is negative to determine the direction of decrease of the objective function [36].

The main advantages of the second-order Newton's method:

(1) if the function is quadratic, then to find the minimum of the objective function $f(x)$, when the preliminary values of the determined parameters are close to the true ones, one iteration is required;

(3) the use of the second partial derivatives in the iterative process allows the increase of the convergence rate, and also to increase the accuracy of the results;

(3) this method is less sensitive to the choice of the initial value of the parameter than the first-order methods.

If the objective function $f(x)$ is not quadratic, then $k$ iterations are required to reach the extremum point until the condition for stopping the iterative process is satisfied [37].

Second-order Newton's method has a number of disadvantages that must be taken into account when implementing it. Calculating the first and second-order derivatives (finite differences) numerically, the accuracy and speed of the method decreases. This is not only due to approximate calculations, but also due to inaccurate approximation of the original objective function. This aspect is especially perceptible in space, around the minimum point, since the first-order derivatives become rather small quantities. When the objective function is not quadratic, the iterative process can loop. As mentioned above, it is necessary at each iteration to check the positive definiteness of the Hessian matrix, since this is the main condition for the convergence of the method. The sign of the Hessian matrix is checked by the Sylvester criterion. The complexity of setting the initial parameter when the function is defined to a small extent (lack of initial data). The need to calculate the second partial derivatives of the function to be minimized. As stated above, the second-order Newton's method was not used in geodesy and surveying due to the complexity of its execution (at that time, the impossibility of complete automation of the computational process).

The convergence rate of Newton's method in the vicinity of a strictly local minimum point is very high (quadratic). The method will not work if the Hessian matrix is degenerate (the determinant of the matrix is zero), and this method may also diverge [38].

The high rate of convergence of Newton's method can be explained by the fact that the quadratic trinomial (2), constructed by taking into account information about both the first and the second derivatives of the objective function, approximates a convex twice differentiable nonlinear function with high accuracy in a sufficiently small neighborhood of this point.

The process of finding the optimal solution using nonlinear programming methods has an iterative nature, which means that, with an increase in the number of iterations, the probability of arriving at the correct solution increases. An important element of the correct operation of all iterative methods is the criterion (rule) for stopping the computational process. It is this criterion that sets the accuracy (from the point of view of mathematics, not geodesy) of achieving a solution, as well as the effectiveness of the method and the amount of computation.

The following stopping criteria are most common in optimization theory:

1. By the absolute value of the difference between the subsequent and previous values of the determined parameter (7):

$$|x_{k+1} - x_k| \leq \varepsilon. \tag{7}$$

2. By the absolute value of the difference between the values of the objective function, the next and the previous iteration (8):

$$|f(x_{k+1}) - f(x_k)| \leq \varepsilon. \tag{8}$$

3. By the absolute value of the derivative of the objective function at the current iteration (9):

$$\left| \frac{\partial f(x)}{\partial x} \leq \varepsilon \right|. \tag{9}$$

Using only one of the criteria can lead to a "false" decision; therefore, it is recommended to take into account several installation criteria in the software algorithm. In all three criteria, the values are less than a known number $\varepsilon$. The user sets this number himself /herself, based on practical experience in solving problems, or after calculating it using formulas.

In this work, the authors analyze the data obtained using methods of the first and second-order, which in the iterative process use derivatives of various orders. Therefore, the authors consider it necessary to note in the work the methods allowing the calculation of derivatives of various orders.

One way to calculate derivatives is by numerical differentiation. Mathematicians turn to it when calculating derivatives of functions given in a table or direct differentiation is difficult. The latter, for example, arises in the case of a complex analytical form of a function. Then, the derivative is interpolated. To calculate the first-order derivative, the Formula (10) can be used:

$$\frac{\delta f(x_1,\ldots,x_n)}{\delta x_n} = \frac{f(x_1+h,\ldots,x_n) - f(x_1-h,\ldots,x_n)}{2 \cdot h}, \tag{10}$$

where $\frac{\delta f(x_1,\ldots,x_n)}{\delta x_n}$ is the first derivative of the objective function $f(x_1,\ldots,x_n)$ with respect to the parameter $x_n$; $h$ is a small increment to the objective function argument.

The increment value $h$ affects the accuracy of the resulting derivative value and the amount of computation. If selecting a very small $h$ round-off error, when calculating with a computer, it can be comparable to or greater than $h$. An algorithm that reduces the error in calculating the derivative is represented by Formula (10). Formula (10) is called the central difference scheme and, according to [33], is the best way to calculate the first-order derivative.

By analogy with obtaining a difference scheme for calculating the first derivative, one can obtain a formula for calculating the second-order derivative of the objective function. The formula for calculating the second-order derivative has the Formula (11):

$$\frac{\delta f(x_1,\ldots,x_n)}{\delta x_n \delta x_n} = \frac{f(x_1+h,\ldots,x_n) - 2f(x_1,\ldots,x_n) + f(x_1-h,\ldots,x_n)}{h^2}, \tag{11}$$

where $\frac{\delta f(x_1,\ldots,x_n)}{\delta x_n \delta x_n}$ is the second-order derivative of the objective function $f(x_1,\ldots,x_n)$ with respect to parameter $x_n$.

In the article, the second-order Newton's method is compared with the conjugate gradient method (the first-order method). This method was chosen because it is the most common in geodetic practice and was used by such well-known surveyors as A.V. Zubov. [39,40], V.A. Kougia [37,41], B.T. Mazurov [42], S.G. Shnitko [43] and other Russian [1,44,45] and foreign [46,47] specialists. The main advantage is that the algorithm does not use second derivatives.

The conjugate gradient method is a kind of continuation of the development of the steepest descent method, which combines two concepts: the gradient of the objective function and the conjugate direction of vectors. The main iterative formula of the method is written in the Formula (12):

$$x_{k+1} = x_k - \lambda_k \cdot P_k, \tag{12}$$

where $P_k$ is the unit vector of conjugate directions; $\lambda_k$ is the length of the movement step at each iteration.

At the zero iteration, the unit vector $P_k$ is taken to be equal to $P_0 = \nabla f(x_1,\ldots,x_n)$. In subsequent calculations, the vector $P_k$ can be calculated using the Formula (13):

$$P_k = \nabla f(x_1,\ldots,x_n)_k + \beta_k \cdot P_{k-1}, \tag{13}$$

where $\beta_k$ is the weighting factor that is used to determine the conjugate directions.

The weighting factor $\beta_k$ can be determined using the Fletcher–Reeves Formula (14):

$$\beta_k = \frac{|\nabla f(x_1,\ldots,x_n)_k|^2}{|\nabla f(x_1,\ldots,x_n)_{k-1}|^2}. \tag{14}$$

According to the presented formulas, the new conjugate direction is obtained by adding the antigradient at the turning point and the previous direction of movement, multiplied by a coefficient $\beta_k$. Thus, the conjugate gradient method creates a search direction, to the optimal value using the information about the search obtained at the previous stages of the descent.

It is worth noting that the works [27,36] noted the benefit of restarting the algorithmic procedure every $n + 1$ steps ($n$ is the number of parameters to be determined). A restart of the computational procedure is necessary in order to "erase" the last direction of the search and start the search algorithm anew in the direction of the fastest descent.

As noted above, the value of step $\lambda_k$ affects the performance of the method. The step size in the conjugate gradient method is selected from the condition of the minimum objective function in the direction of motion, that is, as a result of solving the problem of one-dimensional optimization in the direction of the antigradient.

*2.2. Geodetic Data for Solving the Task*

As mentioned above, the second-order Newton's method was applied to equalize the trilateration network; the network configuration is shown in Figure 1.

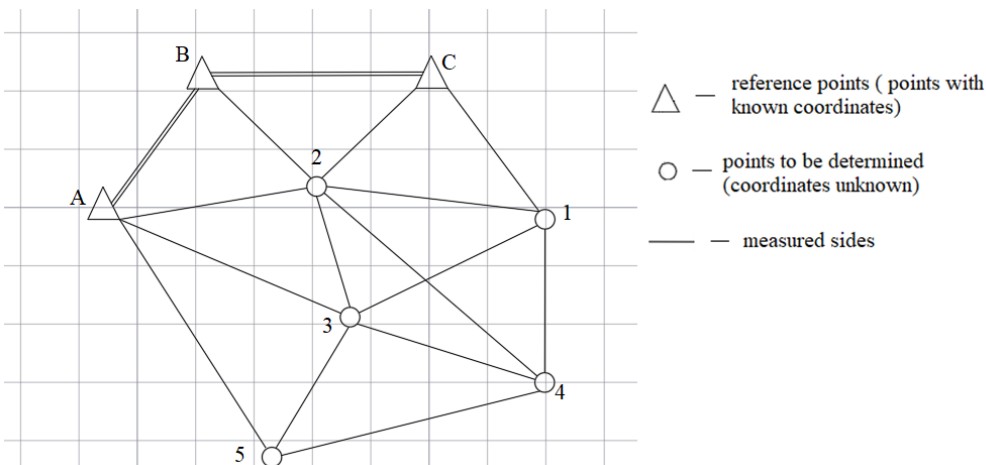

**Figure 1.** Network configuration.

The purpose of solving the problem is to calculate the plane coordinates of points 1–5. To determine the coordinates of the points, an objective function was compiled, with the restrictive condition for minimizing the sum of the squares of the Formula (15):

$$f(z) = \sum_{i=1}^{n} \left[ p_i \cdot (S_{c_i} - S_{m_i})^2 \right], \tag{15}$$

where $n$ is the number of measured distances between points, $p_i$ is weights of the measured sides, $S_{c_i}$ is the vector of calculated distances, $S_{m_i}$ is the vector of measured distances, $z$ is the objective function argument.

Vector $S_{c_i}$ elements are calculated by the Formula (16):

$$S_c = \sqrt{(X_E - X_S)^2 + (Y_E - Y_S)^2}, \tag{16}$$

where $X_E, Y_E$ are the coordinates of the end point of the side, $X_S, Y_S$ are the coordinates of the starting point of the side.

Traditionally, surveying and geodetic networks are adjusted using a parametric method. The essence of this method is:

(1) drawing up parametric communication equations;
(2) linearization of these equations by expanding into a Taylor series taking into account only first-order derivatives;
(3) solution of the obtained systems of equations based on the least squares method.

As can be seen, the traditional method of equalization does not allow the use of objective functions other than the least squares method. Using nonlinear programming

methods, such a possibility appears. Therefore, the network equalization (Figure 1) was also performed using the objective function, which is the minimum of the sum of the modules of the distance corrections. This objective function is expressed by the Formula (17):

$$f(z) = \sum_{i=1}^{n} [p_i \cdot |S_{c_i} - S_{m_i}|]. \tag{17}$$

The coordinates of the starting points are presented in Table 1.

**Table 1.** The coordinates of the starting points.

| Item | Coordinates | |
| --- | --- | --- |
| | **X, m** | **Y, m** |
| A | 645.112 | 426.229 |
| B | 1028.568 | 857.277 |
| C | 740.339 | 1333.496 |

The lengths of lines are presented in Table 2. Due to the uniformity of the measurements, the weights $p_i$ of all measured sides were taken as equal to one.

**Table 2.** Line lengths.

| No. | Line Name | Length, m |
| --- | --- | --- |
| 1 | C–2 | 492.886 |
| 2 | B–2 | 448.178 |
| 3 | A–2 | 445.726 |
| 4 | A–3 | 512.201 |
| 5 | 3–2 | 504.961 |
| 6 | 2–4 | 733.414 |
| 7 | 2–1 | 523.911 |
| 8 | 1–C | 534.601 |
| 9 | 3–1 | 654.977 |
| 10 | 3–4 | 482.249 |
| 11 | 4–1 | 456.648 |
| 12 | 5–A | 617.706 |
| 13 | 5–3 | 322.978 |
| 14 | 5–4 | 700.240 |

## 3. Results

Applying the second-order Newton's method to find the minimum of the objective function (5), the search process was carried out in the MathCAD15 environment. A part of the program is shown in Figure 2.

The preliminary values of the coordinates of the points, as well as the data obtained during the iterative process when using two methods, are presented in Table 3.

The values of the preliminary coordinates were taken specifically far from the minimum point of the objective function (15). This was done to test the main advantage of the second-order Newton's method, namely, the small dependence of the convergence of the method on the preliminary values of the determined parameters, in comparison with gradient methods. The preliminary values of the parameters are presented in Table 4. Table 4 also presents the data obtained during the study.

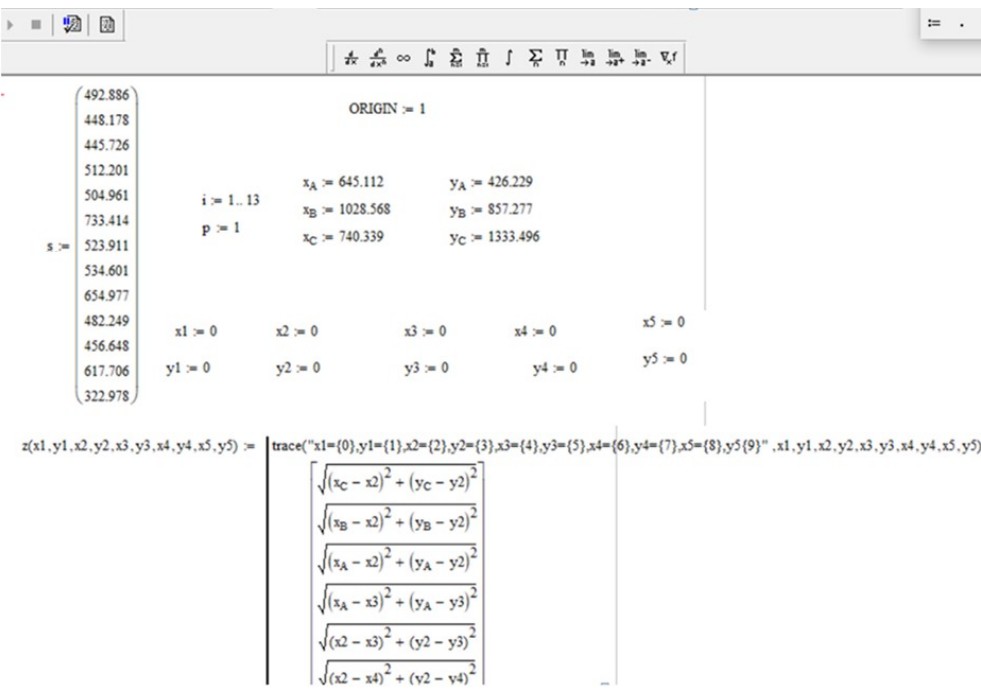

**Figure 2.** Implementation of the second-order Newton's method in MathCAD15.

**Table 3.** Data for comparing the two methods with close setting of preliminary parameter values (objective function by the method of least squares).

| Item | Preliminary Coordinates | | Calculated Coordinates | | | |
|---|---|---|---|---|---|---|
| | | | Second-Order Newton's Method | | Conjugate Gradient Method | |
| | *X*, m | *Y*, m | *X*, m | *Y*, m | *X*, m | *Y*, m |
| 1 | 210.000 | 1235.000 | 213.736 | 1241.368 | 213.763 | 1241.430 |
| 2 | 575.000 | 860.000 | 580.501 | 867.247 | 580.515 | 867.261 |
| 3 | 150.000 | 580.000 | 159.346 | 588.653 | 159.363 | 588.730 |
| 4 | −135.000 | 950.000 | −146.870 | 961.206 | −146.851 | 961.283 |
| 5 | 40.000 | 285.000 | 43.240 | 287.266 | 43.042 | 287.478 |
| NoI [1] | | | 3 | | 389 | |
| OF [2] | 859.468 | | $1.318 \times 10^{-7}$ | | $2.769 \times 10^{-2}$ | |
| CT [3] | | | 25.5 s | | 48.8 s | |

[1] Number of iterations. [2] Objective function value. [3] Computational time.

**Table 4.** Data for comparing the two methods with rough presetting (objective function according to the method of least squares).

| Item | Preliminary Coordinates | | Calculated Coordinates | | | |
|---|---|---|---|---|---|---|
| | | | Second-Order Newton's Method | | Conjugate Gradient Method | |
| | *X*, m | *Y*, m | *X*, m | *Y*, m | *X*, m | *Y*, m |
| 1 | 10.000 | 10.000 | 213.737 | 1241.370 | 213.517 | 1242.222 |
| 2 | 10.000 | 10.000 | 580.501 | 867.248 | 580.685 | 867.828 |
| 3 | 10.000 | 10.000 | 159.347 | 588.656 | 159.880 | 589.763 |
| 4 | 10.000 | 10.000 | −146.869 | 961.209 | −146.365 | 962.073 |
| 5 | 10.000 | 10.000 | 43.237 | 287.272 | 43.290 | 289.190 |
| NoI [1] | | | 11 | | 586 | |
| OF [2] | 859.468 | | $2.245 \times 10^{-5}$ | | 2.413 | |
| CT [3] | | | 42.5 s | | 59.8 s | |

[1] Number of iterations. [2] Objective function value. [3] Computational time.

The use of nonlinear programming methods makes it possible to perform equalization, not only using the objective function of the least squares method (15) but also using the objective function of the least modulus method (17). The data obtained using the new objective function are presented in Table 5.

**Table 5.** Data for comparing the two methods with precise presetting values (objective function according to the method of least modulus).

| Item | Preliminary Coordinates | | Calculated Coordinates | | | |
| | | | Second-Order Newton's Method | | Conjugate Gradient Method | |
| | *X*, m | *Y*, m | *X*, m | *Y*, m | *X*, m | *Y*, m |
|---|---|---|---|---|---|---|
| 1 | 210.000 | 1235.000 | 213.736 | 1241.367 | 213.762 | 1241.379 |
| 2 | 575.000 | 860.000 | 580.500 | 867.246 | 580.515 | 867.246 |
| 3 | 150.000 | 580.000 | 159.346 | 588.652 | 159.352 | 589.665 |
| 4 | −135.000 | 950.000 | −146.869 | 961.205 | −146.406 | 961.591 |
| 5 | 40.000 | 285.000 | 43.231 | 287.275 | 43.300 | 289.085 |
| NoI [1] | | | 103 | | 1233 | |
| OF [2] | 859.468 | | $1.047 \times 10^{-3}$ | | 1.025 | |
| CT [3] | | | 49.1 s | | 100.1 s | |

[1] Number of iterations. [2] Objective function value. [3] Computational time.

## 4. Discussion

As can be seen from Table 3, the coordinates of the determined points of the network were obtained in three approximations, using the second-order Newton's method. The criterion for stopping the search process was the value $\varepsilon$. When solving the problem $\varepsilon$, it was taken as equal to 0.001 m. Figure 3 shows a simplified visualization of an iterative process that was performed using two methods. After analyzing Table 3 and Figure 3, we can conclude that Newton's method is the most efficient in comparison with the gradient method.

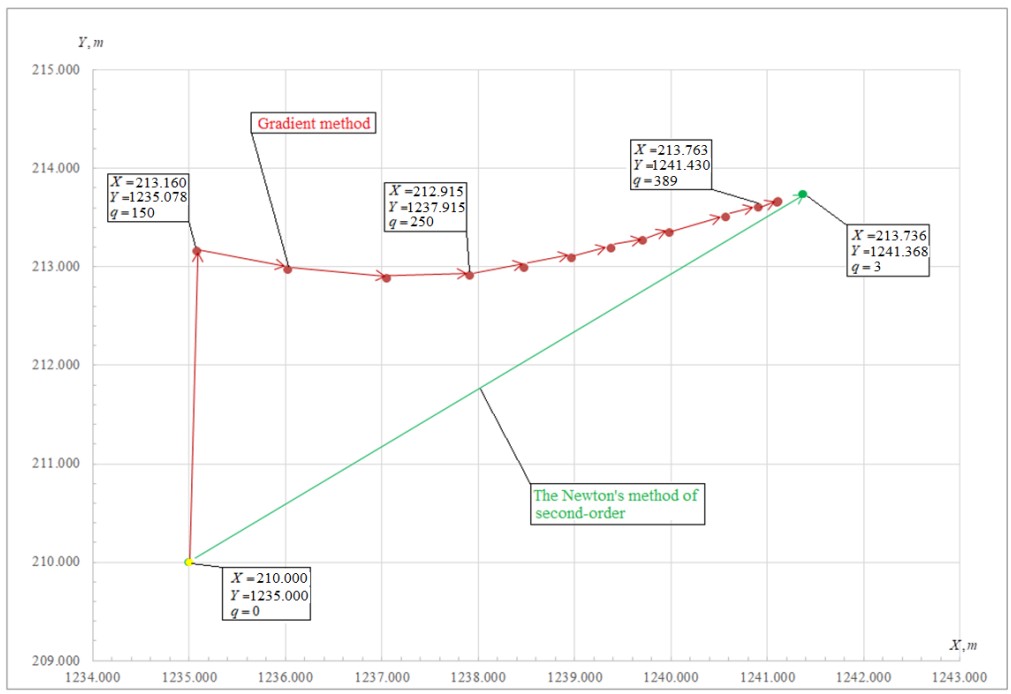

**Figure 3.** Implementation of the second-order Newton's method in MathCAD15.

When using Newton's method, the iterative process is not built according to a linear law, as in the gradient method. The use of second-order partial derivatives allows us to

talk about a quadratic approximation of the objective function, which in turn reduces the number of iterations. From Figure 3, it can be seen that, when approaching the minimum point of function (15), the size of the iteration step in the gradient method decreases, which in turn increases the number of calculations.

According to the data presented in Table 3, it can be seen that the values of the coordinates of the points calculated by two different methods differ. This can be explained by the fact that, in the course of linearization according to Newton's method, the second partial derivatives of the function are used, which are responsible for the concavity of the function and allow the smallest value along the curve to be found. While, for the gradient method to work, only the values of the first derivatives are required (geometrically, this is a tangent), so the minimum value can only be found at the ends of a straight line, not along it.

As mentioned above, the main advantage of Newton's method is the lesser dependence of the convergence of the method on the choice of preliminary values of the sought parameters, in comparison with the gradient methods. The coordinates of the network points were calculated using preliminary values that were set specifically far from the minimum point. Table 4 shows the data obtained during the iterative process. The number of iterations has increased for two methods, but the Newton's method has an order of magnitude less than the gradient method. The discrepancy (more than 100 m) of the preliminary values and the values of the obtained coordinates still affected the accuracy of obtaining the latter, since the value of the objective function increased.

The use of nonlinear programming methods made it possible to calculate the coordinates of points not only using the objective function (15), but also using the objective function of the method of least modules (17). The data obtained during the execution of the iterative process are presented in Table 5. It is worth noting that, despite the close location of the preliminary values to the minimum point, the number of iterations increased in the two methods, compared to the options when the objective function was used (15). The function values have also increased. Analyzing the data presented in Table 5, we can say that, using the gradient method, most likely the point of a local minimum was determined, since the value of the objective function is large enough.

Today it is necessary to develop an algorithm, the use of which allows the user to obtain a correct answer with high accuracy in a short period of time and without taking into account the influence of the preliminary values of the determined parameters. The second-order Newton's method has such resources, due to the use of the second partial derivatives of the objective function, the speed of solving the problem is higher, with a smaller number of approximations compared to the methods of the first-order.

However, in the course of a computational experiment, it was found that this method does not give the correct solution for all preliminary values of the parameters, sometimes the method simply does not work. This is primarily due to the fact that the Hessian matrix indicates the direction of decreasing the function, only if it is positive definite. Therefore, the user needs to prepare the problem for the solution, namely, to calculate the preliminary values, taking into account that they do not make the Hessian matrix negative. If this condition is not followed, then the method may diverge and the method loses its main advantage—the speed of the solution. Using only direct search methods expands the range of selection of preliminary values, since these methods have no restrictions on the sign of derivatives, since derivatives are not used in the iterative process; however, it is necessary to set more conditions for calculating different values of the objective function, which complicates the search process and increases the search time.

The authors of the article propose the creation of a software algorithm based on the second-order Newton's method and on direct search methods, in particular the Powell method and the Davis–Sven–Kempy (DSK) method. The use of this software algorithm will enhance the positive aspects of the second-order Newton's method, namely, to reduce the dependence on the preliminary values of the determined parameters. It would be convenient for the user to use an algorithm in which the number of iterations does not

depend on the preliminary values of the parameters being determined. The main reason for combining the second-order Newton's method with direct search methods is to increase the potential of the method in terms of increasing the speed of the optimization process. A combination of direct search methods, namely the DSK method and the Powell method, was used to create a modified second-order Newton's method.

The essence of the algorithm based on a combination of the DSK method and the Powell method is as follows [48]:

1.  The objective function $F(x^1, x^2, \ldots, x^n)$, depending on the parameters $x^1, x^2, \ldots, x^n$ to be determined, is set.
2.  The preliminary value of the parameter $x^1$ and the increment step $\Delta x_1$ are set.
3.  The increment $\Delta x_1$ is added and subtracted only to the first parameter $x^1$, the rest of the parameters are also given preliminary values, but they remain unchanged.
4.  The values of the objective function are calculated with the changed parameters $F(x^1 - \Delta x_1, x^2, \ldots, x^n)$ and $F(x^1 + \Delta x_1, x^2, \ldots, x^n)$
5.  The new value of the determined parameter is calculated by the Formula (18):

$$x^{1*} = x^1 + \frac{\Delta x_1 (F(x^1 - \Delta x_1, x^2, \ldots, x^n) - F(x^1 + \Delta x_1, x^2, \ldots, x^n))}{2 \cdot (F(x^1 - \Delta x_1, x^2, \ldots, x^n) - F(x^1, x^2, \ldots, x^n) + F(x^1 + \Delta x_1, x^2, \ldots, x^n))}. \quad (18)$$

6.  The next parameter $x^2$ is changed and the new value of the function is calculated, only the value $x^{1*}$ is substituted into the target function instead of the parameter $x^1$.

In general, this method may require a sufficiently large number of iterations to find the optimal solution. However, its main advantage is that its solution area is much larger in comparison with the classical second-order Newton's method.

The authors of the article have developed the following algorithm to minimize the main disadvantages of the Newton's method algorithm:

1.  Step 1: The user creates an objective function $F(x^1, x^2, \ldots, x^n)$ and chooses with what constraint he/she will find the minimum of the objective function (by the method of least squares or by the method of least modules); it is recommended to use the least squares method for solving geodetic tasks;
2.  Step 2: Sets any preliminary values of the parameters to be determined (it is recommended to set either previously known to true values or accept all parameters as equal to zero);
3.  Step 3: using the methods of quadratic approximation, namely the Powell–DSK method, in two approximations, the preliminary values are refined according to Formula (18);
4.  Step 4: The Hessian matrix is created, and its positiveness is checked; if the condition is met, then the revised preliminary values are used in the next step. If the Hessian matrix is not positive, then step 3 is performed again;
5.  Step 5: The obtained refined preliminary values are used in the second-order Newton's method, the matrix of the first derivatives and the matrix of the second derivatives are formed;
6.  Step 6: An iterative process is performed according to Formula (5) until the stopping criterion is met (Formula (7)). The stopping criterion is chosen by the user;
7.  Step 7: The accuracy of the obtained parameter values is evaluated. To estimate the accuracy of the obtained parameters, an inverse weight matrix is used.

When performing the equalization of surveying and geodetic measurements using nonlinear programming methods, difficulties arise in performing the accuracy assessment, since the iterative process of the first and second-order methods does not require compiling a matrix of normal equations of unknowns; therefore, it is not possible to find the inverse weight matrix $Q$. The assessment of the accuracy of the data obtained during the use of nonlinear programming methods was given attention in the works of G.V. Makarov. [49], V.I. Mitskevich [35,50–52], Ch.N. Zheltko [41]. V.I. Mitskevich notes that a fragment of the

inverse matrix of weights can be obtained by the generalized method of G.V. Makarov in [49,53].

The procedure for evaluating the accuracy of the coordinates of the points of the mine surveying and geodetic network obtained by nonlinear programming methods is given in [54–59]. The accuracy estimation algorithm using nonlinear programming methods is described in detail in [60]. V.I. Mitskevich in his works [50,52,60] asserts that, when optimizing using the objective function of the least squares method according to Newton's second-order method, to compose the inverse weight matrix, one can use the Hessian matrix according to Formula (19):

$$Q = 2H^{-1} = 2 \cdot \begin{pmatrix} \frac{\partial^2 f(x^1,...,x^n)}{\partial x^1 \partial x^1} & \frac{\partial^2 f(x^1,...,x^n)}{\partial x^1 \partial x^2} & \cdots & \frac{\partial^2 f(x^1,...,x^n)}{\partial x^1 \partial x^n} \\ \frac{\partial^2 f(x^1,...,x^n)}{\partial x^2 \partial x^1} & \frac{\partial^2 f(x^1,...,x^n)}{\partial x^2 \partial x^2} & \cdots & \frac{\partial^2 f(x^1,...,x^n)}{\partial x^2 \partial x^n} \\ \cdots & \cdots & \ddots & \\ \frac{\partial^2 f(x^1,...,x^n)}{\partial x^n \partial x^1} & \frac{\partial^2 f(x^1,...,x^n)}{\partial x^n \partial x^1} & \cdots & \frac{\partial^2 f(x^1,...,x^n)}{\partial x^n \partial x^n} \end{pmatrix}^{-1}. \quad (19)$$

In the article, to assess the accuracy of the obtained values of the parameters determined by the second-order Newton's method, the inverse weight matrix will be compiled according to Formula (19).

The modified Newton's method was applied with preliminary values of the coordinates of the determined points, at which the classical second-order Newton's method does not work, since it diverges. The data obtained during the iterative process are presented in Table 6. Table 6 also presents data that make it possible to assess the accuracy of the obtained coordinate values, namely, the root-mean-square errors of the coordinates of the item being determined and the mean-square error of the unit of weight.

**Table 6.** The data obtained during the equalization of the trilateration network using the modified second-order Newton's method.

| | Item | Preliminary Coordinates | | Calculated Coordinates Modified Second-Order Newton's Method | |
|---|---|---|---|---|---|
| | | X, m | Y, m | X, m | Y, m |
| | 1 | 0.000 | 0.000 | 213.736 | 1241.368 |
| | 2 | 0.000 | 0.000 | 580.501 | 867.247 |
| | 3 | 0.000 | 0.000 | 159.346 | 589.653 |
| | 4 | 0.000 | 0.000 | −146.870 | 961.206 |
| | 5 | 0.000 | 0.000 | 43.240 | 287.266 |
| | NoI [1] | | | 28 | |
| | OF [2] | | | $5.730 \times 10^{-4}$ | |
| | CT [3] | | | 98.7 s | |
| RS [4] | $m_{x_1}/m_{y_1}$, mm | | | $1.047 \times 10^{-3}/5.180 \times 10^{-4}$ | |
| | $m_{x_2}/m_{y_2}$, mm | | | $4.080 \times 10^{-4}/3.980 \times 10^{-4}$ | |
| | $m_{x_3}/m_{y_3}$, mm | | | $5.090 \times 10^{-4}/5.300 \times 10^{-4}$ | |
| | $m_{x_4}/m_{y_4}$, mm | | | $5.400 \times 10^{-4}/6.870 \times 10^{-4}$ | |
| | $m_{x_5}/m_{y_5}$, mm | | | $6.56 \times 10^{-4}/7.84 \times 10^{-4}$ | |

[1] Number of iterations. [2] Objective function value. [3] Root mean square errors of coordinates of the determined point. [4] Computational time.

The Modified second-order Newton method should also be compared with the Broyden–Fletcher–Goldfarb–Shanno method (BFGS). It should be noted that, in contrast to the classical second-order Newton method, the Hessian matrix is not calculated directly in quasi-Newtonian methods, that is, there is no need to find second-order partial derivatives. Instead, the Hessian is calculated approximately from the previous approximations. One of the most effective quasi-Newtonian methods is the BFGS method; the advantage of this algorithm is the simplicity of software implementation. However, the main disadvantage of using this method is the increase in the number of iterations to find the minimum of the

objective function. This disadvantage can be mitigated by the fact that, due to the performance of modern computers when solving simple optimization problems, the increase in the number of approximations does not become noticeable in time for the user. The data obtained during the iterative process are presented in Table 7.

**Table 7.** The data obtained during the equalization of the trilateration network using the modified second-order Newton's method and the Broyden–Fletcher–Goldfarb–Shanno method.

| Item | Preliminary Coordinates | | Calculated Coordinates | | | |
| | | | Second-Order Newton's Method | | BFGS | |
| | *X*, m | *Y*, m | *X*, m | *Y*, m | *X*, m | *Y*, m |
|---|---|---|---|---|---|---|
| 1 | 0.000 | 0.000 | 213.736 | 1241.368 | 288.806 | 1070.175 |
| 2 | 0.000 | 0.000 | 580.501 | 867.247 | 652.085 | 777.799 |
| 3 | 0.000 | 0.000 | 159.346 | 589.653 | 201.616 | 380.299 |
| 4 | 0.000 | 0.000 | −146.870 | 961.206 | −63.580 | 810.951 |
| 5 | 0.000 | 0.000 | 43.240 | 287.266 | 71.944 | 39.043 |
| NoI [1] | | | 28 | | 144 | |
| OF [2] | 859.468 | | $5.730 \times 10^{-4}$ | | $44.863 \times 10^{3}$ | |
| CT [3] | | | 98.7 s | | 122.1 s | |

[1] Number of iterations. [2] Objective function value. [3] Computational time.

## 5. Conclusions

It should be noted that the methods that use derivatives of higher orders in the search process have a wide range of applications in geodesy and surveying. To test the possibility of implementing the second-order Newton's method in surveying and geodetic production, the trilateration network was equalized. In the course of solving this problem, the main advantages of the method were confirmed; namely, a high convergence rate (compared to methods using first-order derivatives) and the possibility of using rough values of preliminary parameters for an iterative process. On the other hand, the main disadvantages of the method were also confirmed: it is a highly complex computational process (formation of the Hessian matrix and control of its sign). To reduce the influence of the disadvantages of this method, a software algorithm was created based on the second-order Newton's method and on direct search methods, in particular the Powell method and the Davis–Sven–Kempy (DSK) method. The use of this software algorithm will enhance the positive aspects of the second-order Newton's method; namely, to reduce the dependence on the preliminary values of the determined parameters. It would be convenient for the user to use an algorithm in which the number of iterations does not depend on the preliminary values of the parameters being determined. The main reason for combining the second-order Newton's method with direct search methods is to increase the potential of the method in terms of increasing the speed of the optimization process. The prospect of further research is to expand the range of problems to be solved by a modified second-order Newton's method and to study its efficiency and productivity in new conditions.

**Author Contributions:** Conceptualization, investigation, methodology, and software, D.B. Formal analysis, writing—review and editing, M.M. All authors have read and agreed to the published version of the manuscript.

**Funding:** The research was carried out at the expense of a subsidy for the implementation of the state task in the field of scientific activity for 2021 № FSRW-2020-0014.

**Data Availability Statement:** Data sharing is not applicable to this article.

**Conflicts of Interest:** The authors declare no conflict of interest.

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
