# Peer review of "Adjustment of Planned Surveying and Geodetic Networks Using Second-Order Nonlinear Programming Methods"

_computation, doi:10.3390/computation9120131_

Round 1

Author Response

An Article " Adjustment of Planned Surveying and Geodetic Networks  Using Second-Order Nonlinear Programming Methods". The authors thank the reviewer for their careful reading of the work and observations made. The answer to the comment is contained in the attached file.

Reviewer 2 Report

The authors use an algorithm based on the Newton method and
on the direct search methods Powell and Davis-Sven-Kempy
for geodetic networks.

Remarks

1) The algorithm have to be presented with details: step 1, step 2 ...

2) line 462. The number of the formula should be (19) in place of (14)
Q=2H^{-1}=H. The inverse of the Hessian matrix is half of the Hessian matrix ?

3) lines 186-187.
``(3) this method is less sensitive to the choice of the initial value of
the parameter than the first-order methods.''

I do not agree. Generally, the initial guest for Newton method must be
near the solution. Can you show an example for your affirmation (3)?

4) The Hessian matrix is computed with finite differences methods and the
positive definite is tested by the Sylvester criterion which is not
efficient for problems with large number of parameters.
What is the computational time for a problem with 100 parameters?

5) The method presented should be compared with Davidon-Fletcher-Powell or
Broyden-Fletcher-Goldfarb-Shanno methods.

Author Response

An Article " Adjustment of Planned Surveying and Geodetic Networks  Using Second-Order Nonlinear Programming Methods". The authors thank the reviewer for their careful reading of the work and observations made.The answer to the comment is contained in the attached file

Round 2

Reviewer 1 Report

The authors have satisfactorily addressed all my comments and concerns raised in the previous round. Therefore, I recommend publication of the paper in the current form.

Reviewer 2 Report

1) formula (14)
Q=2H^{-1}=2(...)^{-1}

2) Table 7. For an optimization problem with n > 20,
I think that the computational time for BFGS is smaller.